# OpenReview forum: "AJF: Adaptive Jailbreak Framework Based on the Comprehension Ability of Black-Box Large Language Models"
_ICLR.cc/2026/Conference — Submitted to ICLR 2026_

### Official Review · Reviewer_XM8h · 2025-10-28

**Soundness:** 1
**Presentation:** 1
**Contribution:** 1
**Rating:** 0
**Confidence:** 5

**Summary:**

This paper proposes AJF, a capability-aware jailbreak attack strategy against black-box LLMs. By categorizing models into Type-I and Type-II based on comprehension ability, it applies customized encryption and mutation techniques to bypass input, inference, and output-stage defenses.

**Strengths:**

This paper introduces a model categorization framework that classifies aligned LLMs into two distinct types based on their comprehension ability, providing useful insights into model vulnerability patterns and guiding adaptive jailbreak strategies.

**Weaknesses:**

1. The novelty is limited. The core components are just the extentions if the existing work.
2. The classification of LLMs into Type-I and Type-II is based on a single Caesar cipher test, which is heuristic and may not reflect real comprehension capability.
3. The attack prompt templates are deterministic and structurally repetitive, which makes them potentially vulnerable to static pattern-based defenses.
4. The framework is only tested under static conditions and does not assess resilience against adaptive defenses, fine-tuning, or dynamic moderation strategies.
5. The paper compares AJF with only a limited set of baselines.

**Questions:**

Please refer to the weaknesses.

---

> ### Author Response · Authors · 2025-11-19
> **Thank you for your comment, and I look forward to further discussion with you.**
>
> Regarding Weakness 1:
> Thank you for your comment. However, I respectfully disagree with the claim that the novelty of our work is limited. Prior studies that perform jailbreak attacks by shuffling harmful prompts—such as
> CodeChameleon (Huijie Lv, et al. Codechameleon: Personalized encryption framework for jailbreaking large language models. arXiv preprint arXiv:2402.16717, 2024.),
> CipherChat (Yuan, Youliang, et al. "GPT-4 Is Too Smart To Be Safe: Stealthy Chat with LLMs via Cipher." ICLR. 2024.), and
> FlipAttack (Yue Liu, et al. Flipattack: Jailbreak llms via flipping. ICML, 2025.)—
> indeed share certain conceptual similarities, but each of them introduces its own innovations. Our work is similar in spirit, yet the core ideas and contributions are fundamentally different in the following aspects:
>
> 1.We design adaptive jailbreak strategies based on the target model’s comprehension ability, whereas previous methods never consider the model’s understanding capacity before the attack. As a result, their attack strategies may not match the inherent comprehension ability of the model, leading to suboptimal performance. In contrast, our method explicitly adapts the attack design to different model comprehension levels.
>
> 2.Our jailbreak strategy requires the coordinated use of Mu, En_prompt, and a distinct encryption strategy En_response.
> We demonstrate that if En_prompt and En_response use the same encryption mechanism, the jailbreak success rate drops significantly. The high success rate shown in our experiments is achieved because these three components work together. This coordinated strategy has not been explored in prior work.
>
> On the same datasets, our method surpasses CodeChameleon, CipherChat, and FlipAttack in jailbreak success rates on GPT-4 and GPT-4o.
> These empirical improvements further support that our approach is not a simple extension but a meaningful advancement beyond existing methods.
>
> For these reasons, we believe our work presents clear and substantive novelty.
>
> Regarding Weakness 2: We sincerely apologize for the misunderstanding. Due to space limitations, we were unable to provide a more detailed explanation of the underlying principles of the probe prompt. Please allow me to clarify here: In our paper, “comprehension ability” refers specifically to whether a model can correctly understand and execute the entire operational workflow of MuDeEn. The probing prompt used in the paper replaces the initial harmful query with a harmless execution task within the MuDeEn pipeline. If a model can successfully complete this task, it indicates that the model is capable of understanding all steps required by MuDeEn, and thus can be attacked using the MuDeEn strategy. Therefore, additional tests for general language understanding are not necessary for our purpose. If you believe this clarification is essential, we will incorporate it into a revised version of the paper.
>
> Regarding Weakness 3 and 4:
> Thank you for your comment. However, I must respectfully disagree with your viewpoint. Below are my explanations and thoughts, and I sincerely welcome further discussion:
>
> 1.If an LLM attempts to defend specifically against our proposed strategy, it may lead to a degradation in the model’s usability.
> Our method uses two encryption mechanisms, En_prompt and En_response. These encryption strategies themselves are not designed to jailbreak LLMs. If an LLM begins defending against such encryption patterns, it may fail to perform legitimate encryption-related tasks that are not associated with jailbreak attacks. Moreover, such defensive fine-tuning may trigger ripple effects, causing the LLM to forget or mishandle tasks such as Caesar cipher operations, letter-shifting transformations, or even basic data structures like binary trees. This would result in a significant decline in the LLM’s overall usability.
>
> 2.Current research shows that existing defense methods do not involve fine-tuning LLMs specifically to counter encryption strategies. To the best of our knowledge, there is no evidence demonstrating that LLM defense techniques target or adapt to encryption methods similar to those used in our work.
>
> 3.Our work introduces a combinational strategy that achieves high jailbreak success rates on newer models.
> Even if static defenses were able to block the specific instantiation used in our experiments, one could simply replace En_prompt and En_response with alternative encryption mechanisms and conduct a new attack. The flexibility and recombinability of our strategy implies that defending against one fixed configuration does not neutralize the overall methodology.
>
> Regarding Weakness 5: I am very sorry, but I cannot agree with your assessment. The baselines we selected for comparison are among the most recent and strongest methods in the field, such as FlipAttack (ICML 2025) and CipherChat (ICLR 2024), among others.

---

### Official Review · Reviewer_2YX4 · 2025-10-30

**Soundness:** 3
**Presentation:** 3
**Contribution:** 2
**Rating:** 4
**Confidence:** 4

**Summary:**

This paper presents Adaptive Jailbreak Framework (AJF), a novel method for attacking black-box large language models.

**Strengths:**

- Novel Conceptual Framework: The paper introduces a compelling new perspective by directly linking jailbreak effectiveness to the target model's comprehension ability. The classification of LLMs into Type-I and Type-II, while simple, is a conceptually insightful approach that moves the field beyond one-size-fits-all attacks. This adaptivity represents a more sophisticated paradigm for adversarial attacks on LLMs.
- Significant Implications for LLM Safety: By demonstrating that a model's advanced comprehension can be turned against its own safety mechanisms, this work uncovers a deep and potentially fundamental vulnerability. It provides invaluable insights for the LLM safety and alignment community, suggesting that future defense mechanisms must account for this attack vector. The research serves as a powerful red-teaming contribution, highlighting critical areas for improvement.

**Weaknesses:**

- While the paper introduces a novel "adaptive" perspective, its underlying technical components are largely clever orchestrations of existing primitives rather than fundamental breakthroughs. Programmatic obfuscation via code-like structures and the use of encryption to bypass safety filters are well-established paradigms, explored in prior work such as CodeChameleon and CipherChat. Therefore, the primary contribution lies in the strategic combination of these techniques to form a multi-stage attack, rather than in the invention of a new attack modality itself.
- Furthermore, the framework's central claim to adaptiveness hinges on a model categorization criterion that lacks robustness and generalizability. The classification of LLMs into a rigid Type-I/Type-II binary is based on a single, highly-engineered probe prompt. This approach oversimplifies the multi-dimensional nature of LLM comprehension, which exists on a continuous spectrum, and creates a potential single point of failure where misclassification could lead to suboptimal attack strategies. The paper does not address the handling of models on the boundary or validate the criterion's consistency across diverse tasks.
- Finally, the evaluation protocol suffers from significant methodological weaknesses that undermine the reliability of the reported results. The exclusive reliance on an "LLM-as-a-judge" framework introduces a risk of systemic, homologous bias. More critically, the baseline comparisons are not rigorous; many results are directly adopted from previous studies, which likely used different datasets and experimental settings. The absence of controlled, head-to-head experiments prevents a fair and scientifically valid assessment of the method's purported superiority over the state of the art.

**Questions:**

Seen in weakness

---

> ### Author Response · Authors · 2025-11-19
>
> I am truly honored to receive your valuable and insightful review. Please allow me to respond to your comments, and I would be deeply grateful and honored if you were willing to further discuss or exchange ideas with me. My detailed responses are provided below.
>
> Regarding Weakness 1:
>      Thank you for your comments. Since I am familiar with the work of CodeChameleon and CipherChat, please allow me to clarify the differences between our paper and these two approaches, so that the innovations of our work can be better understood.
>
> Comparison with CodeChameleon
> 1. It does not use a reasonable initial mutation strategy.
> CodeChameleon does not perform a proper initial mutation on the harmful prompt before encryption. Directly encrypting a harmful prompt makes the model more likely to treat the decrypted content as a harmful query, thereby recognizing the jailbreak intention and refusing to answer. In contrast, our work explores a Mu strategy that is designed to cooperate with the encryption scheme, and our experiments show that this significantly improves the jailbreak success rate.
> 2. It does not design more complex attacks for Type-II models, nor does it encrypt harmful responses.
> CodeChameleon does not introduce more advanced attacks targeting Type-II models with stronger comprehension abilities, and it does not encrypt harmful responses to bypass output-side safety filters. As a result, its jailbreak success rate is lower than ours.
>
> Comparison with CipherChat
> 1. It also does not use a suitable initial mutation strategy (Mu) together with its encryption method.
> 2. It uses the same simple Caesar cipher for both input and output encryption.
> This simple strategy makes it easier for the model to infer the attack intention itself. Because the encryption scheme is overly simplistic, its jailbreak success rate on the same dataset is even lower than CodeChameleon; this has been discussed and experimentally demonstrated in the CodeChameleon paper.
> 3. It does not design more complex attacks for Type-II models.
> Although CipherChat encrypts the harmful responses to avoid output-side safety detection, it repeatedly uses the same simple encryption mechanism on both the input and output, leading to limited success.
>
> In our experiments, both of these methods achieve lower jailbreak success rates than ours on the same datasets.
>
> Compared with previous works that use encryption for jailbreak attacks, our paper has two essential differences:
> 1. Our method designs different attack strategies based on different model comprehension levels.
> Previous strategies never considered the model’s inherent understanding ability before launching the attack, resulting in a mismatch between the attack strategy and the model’s comprehension capability.
> 2. Our strategy must jointly use Mu, En_prompt, and En_response.
> En_response must use an encryption scheme that is different from En_prompt, and the initial mutation Mu must cooperate with the chosen En_prompt. This Mu must be adjusted when the En_prompt changes. In our paper, the Mu strategy is specifically designed for the En_prompt we propose. Without this cooperation, the success rate drops.
> The paper presents the most effective strategy we have identified: Mu, En_prompt, and En_response must work together to achieve high jailbreak success rates.
> The comparisons under the same datasets can be found in Table 1 and Table 3.
>
> Regarding Weakness 2:
>     We sincerely apologize for the misunderstanding. Due to space limitations, we were unable to provide a more detailed explanation of the underlying principles of the probe prompt. Please allow me to clarify here: In our paper, “comprehension ability” refers specifically to whether a model can correctly understand and execute the entire operational workflow of MuDeEn. The probing prompt used in the paper replaces the initial harmful query with a harmless execution task within the MuDeEn pipeline. If a model can successfully complete this task, it indicates that the model is capable of understanding all steps required by MuDeEn, and thus can be attacked using the MuDeEn strategy. Therefore, additional tests for general language understanding are not necessary for our purpose. If you believe this clarification is essential, we will incorporate it into a revised version of the paper.

---

> ### Author Response · Authors · 2025-11-19
>
> Regarding weakness 3:
> Thank you very much for your valuable comment. We believe that the comparisons presented in our paper are fair, and our reasons are as follows:
>
> 1.Using LLMs as evaluators is a standard practice in jailbreak-related studies. To mitigate the risk of systemic homologous bias, we employed two LLMs that are different from the target model. Both LLMs produced consistent evaluation results. This explanation is provided in Section 4.1.4 of our paper.
>
> 2.We are confident that our comparison protocol is fair. As explained in Section 4, we used the same datasets as those claimed in the original papers.
>
> For each baseline method, our experiments strictly followed the datasets specified in the corresponding publications. Specifically, regarding Table 1, the explanation on dataset fairness is provided in Lines 398–406. Except for GPTFuzzer, all results were taken from CodeChameleon, and our evaluation of our own method also used the dataset claimed by CodeChameleon. For GPTFuzzer, its dataset is a subset of ours; therefore, to ensure fairness, we adopted the best results reported in its paper for comparison. We also explicitly acknowledge that on LLaMA, GPTFuzzer achieves a higher jailbreak success rate than our method.
>
> Regarding Table 3, the explanation on dataset fairness can be found in Lines 419–424. Except for IRIS, all other baseline results came from their original papers, all of which reported experiments conducted on the AdvBench dataset. Our results were evaluated on the same dataset for a fair comparison. Additional details about the IRIS dataset are provided in Lines 425–430.
>
> If you believe that the fairness of dataset usage should be mentioned earlier, we can move this explanation to Section 4.1.1. Due to formatting requirements, we cannot add special annotations within the tables for methods using different datasets; therefore, this information is provided in the main text. If you think this explanation is not sufficiently prominent, we are willing to relocate it to a more noticeable position.
>
> We sincerely appreciate your suggestions and are fully willing to revise the paper accordingly.

---

### Official Review · Reviewer_bZ8G · 2025-11-01

**Soundness:** 3
**Presentation:** 3
**Contribution:** 3
**Rating:** 4
**Confidence:** 3

**Summary:**

This paper introduces the Adaptive Jailbreak Framework (AJF), a novel method for bypassing the safety mechanisms of black-box LLMs. The framework is built on the core insight that the effectiveness of a jailbreak attack should be tailored to the target model's comprehension ability. AJF first classifies LLMs into two categories—Type-I (limited comprehension) and Type-II (strong comprehension)—using a specialized probe prompt. For less capable Type-I models, it employs the MuEn strategy, which combines programmatic prompt mutation with structural encryption to evade input and inference defenses. For more advanced Type-II models, it uses a more sophisticated MuDeEn strategy, which adds a dual-end encryption layer that compels the LLM to generate an encrypted response, thereby circumventing output-level moderation. Through extensive experiments, the authors demonstrate the framework's high efficacy, achieving near-perfect attack success rates of 98.9% on GPT-4o and 99.8% on GPT-4.1. They show that more capable models can be manipulated into executing complex, multi-stage attacks that bypass their own safety filters.

**Strengths:**

* Novel and well-motivated framework: Adapting adversarial attacks to the comprehension level of the target LLM is an insightful and novel contribution to the field of LLM jailbreaking.
* Strong empirical performance: AJF demonstrates state-of-the-art performance, achieving extremely high ASR against some of the most powerful publicly available models.
* The framework is well-designed to circumvent multiple layers of LLM defenses, including input, inference, and output moderation, which explains its high success rate.  The core idea of classifying models into Type-I and Type-II and applying tailored strategies (MuEn vs. MuDeEn) is a sophisticated conceptual contribution that reflects a deeper understanding of the LLM attack surface.
* Efficiency: unlike iterative or query-intensive methods (e.g., fuzzing or optimization-based attacks), AJF is designed as a single-query attack.

**Weaknesses:**

### Oversimplified LLM Categorization:
The framework's foundational step—classifying LLMs into a binary Type-I/Type-II distinction—relies on a single, complex probe prompt, which is a potential single point of failure and may not be robust. ``Comprehension'' is a spectrum, but the framework reduces it to a binary classification based on a single, engineered task (as in Sec. 3.2). The paper does not investigate the consequences of misclassification (e.g., applying the MuDeEn strategy to a Type-I model) or the sensitivity of the probe to small perturbations, making the robustness of this critical step unclear.

### Insufficient Ablation Study to Isolate Component Contributions
The ablation study in Section 4.3, while demonstrating the value of En_response, is too limited to fully disentangle the contributions of all framework components and validate the core adaptive hypothesis.
* Contribution of Mu is Unclear: The study never isolates the effect of the programmatic mutation (Mu). An experiment comparing MuEn (mutation + encryption) against an En-only strategy is missing. This makes it impossible to know how much of the success on Type-I models is due to the mutation versus the encryption.
* Core Hypothesis Not Empirically Tested: The central claim is that strategies must be adapted. The most direct test of this would be to apply the "wrong" strategy (e.g., the complex MuDeEn on a Type-I model) and show that it fails. The paper hypothesizes this would happen (Lines 475-477) but provides no experimental results to prove it.

Overall, I feel that the evaluation section could have been significantly improved to meet the publication quality, in terms of self-containedness and completeness, and extensiveness.

**Questions:**

* Robustness of the categorization probe: The binary Type-I/II classification is foundational to your adaptive framework. Could you comment on the robustness of using a single probe prompt? How sensitive is the classification to minor variations in the prompt's wording or the model's output formatting? Did you consider using a suite of prompts to generate a more continuous capability score, which might offer a more robust basis for strategy selection?

* The central thesis of the paper is that the attack strategy must be matched to the model's capability. To provide direct evidence for this, did you run experiments applying the ``wrong'' strategy? Specifically, what was the performance when applying the complex MuDeEn strategy to a model you classified as Type-I (e.g., Llama2-13b)? A significant drop in performance would provide powerful empirical validation for your core claim.

* In Section 3.4, there appears to be a notational inconsistency. An is first defined as the intermediate natural language answer (Line 261), but then in Equation (6) it is used to represent the final decrypted output. The prose also introduces An* (Line 268). Could you clarify the precise definitions of An and An* and ensure the notation is consistent throughout the section to avoid confusion?

---

> ### Author Response · Authors · 2025-11-19
> **I am truly honored to receive such valuable and insightful comments. We greatly appreciate your feedback and will carefully incorporate your suggestions to achieve a higher-quality revised version of the paper.**
>
> I am truly honored to receive such valuable and insightful comments. We greatly appreciate your feedback and will carefully incorporate your suggestions to achieve a higher-quality revised version of the paper.
>
> Regarding weakness oversimplified LLM categorization and question 1:
>       We sincerely apologize for the misunderstanding. Due to space limitations, we were unable to provide a more detailed explanation of the underlying principles of the probe prompt. Please allow me to clarify here:
>       In our paper, “comprehension ability” refers specifically to whether a model can correctly understand and execute the entire operational workflow of MuDeEn. The probing prompt used in the paper replaces the initial harmful query with a harmless execution task within the MuDeEn pipeline. If a model can successfully complete this task, it indicates that the model is capable of understanding all steps required by MuDeEn, and thus can be attacked using the MuDeEn strategy. Therefore, additional tests for general language understanding are not necessary for our purpose. If you believe this clarification is essential, we will incorporate it into a revised version of the paper.
>
> Regarding weakness contribution of Mu is unclear:
>     Thank you for your insightful comment. Your feedback has been extremely helpful for improving our work. Motivated by your suggestion, we immediately conducted an ablation study to evaluate the effect of the programmatic mutation (Mu) in the MuEn strategy.
>     The results clearly show that including Mu significantly improves the jailbreak success rate. For Llama2-13B, the success rate increases from 75.6% (without Mu) to 94.5% (with Mu). For GPT-4o, the success rate increases from 86.6% (without Mu) to 93.1% (with Mu).
>     We will include these detailed results in a revised version of the paper.
>
> Regarding weakness core hypothesis not empirically tested and question 2:
>     Thank you very much for this helpful comment. Please allow me to offer some clarification.
>     In theory, Type-I models have already been shown—during the classification stage—to be unable to understand the task required by MuDeEn. Therefore, for a Type-I model, using the MuDeEn strategy will fail regardless of whether the requested task is harmful or harmless, because the model cannot execute the workflow that MuDeEn requires.We have conducted experiments to verify this point. The results show that when applying the MuDeEn strategy to a Type-I model, the model does not refuse to answer, but it outputs responses that are completely irrelevant to the harmful query, resulting in a jailbreak success rate of 0%.
>     We will include this conclusion in a revised version of the paper.
>
> Regarding question 3:
>     Thank you for pointing out this issue. We will carefully review the notation used throughout the paper, especially in Section 4.3, to ensure that the definitions of  An and An* are precise and that the notation is correct in the revised version.

---

> ### Author Response · Authors · 2025-11-27
> **Since your feedback is highly valuable, we have seriously considered your suggestions and made the following revisions to the manuscript.**
>
> Since your feedback is highly valuable, we have seriously considered your suggestions and made the following revisions to the manuscript. We hope the new version of the paper is satisfactory and has fully addressed your concerns:
>
> 1.We have revised the relevant descriptions of An and An* in Section 3.4 to clarify the specific definitions of the symbols. We explicitly state that An* is an intermediate response from the jailbroken LLM, not the final answer, and that An is the final result obtained after decryption using the De_response mechanism.
>
> 2.We have provided a clear explanation of the probe prompt in Appendix A.3, elaborating on its design rationale.
>
> 3.We have supplemented the paper with three additional experiments, which are:
>
> 3.1 An attack using only the En strategy against GPT-4o, achieving an Attack Success Rate (ASR) of 86.6%. This result is now presented in Table 4 of the main text.
>
> 3.2 An attack using the MuDeEn strategy against Type-I LLMs, which yielded an ASR of 0%. A detailed explanation is provided in Section A.2.1 of the main text.
>
> 3.3 An experiment using only the En strategy against Llama2-13b, resulting in an ASR of 75.6%, which is lower than the MuEn strategy. This is discussed in detail in Section A.2.2 of the main text.

---

### Official Review · Reviewer_kQ78 · 2025-11-02

**Soundness:** 2
**Presentation:** 2
**Contribution:** 2
**Rating:** 2
**Confidence:** 4

**Summary:**

Adaptive Jailbreak Framework (AJF) provides a prompting strategy to design jailbreak prompts by hiding the malicious task behind sophisticated benign tasks. The main idea being -- when the model focuses on the complex comprehension and decryption, the safety filters are not triggered and the output ends up answering the malicious prompt. The paper demonstrates that this attack is able to successfully attack latest state of the art models.

**Strengths:**

1.The paper highlights an important failure mode of LLMs when trying to solve multiple tasks simultaneously.
2. The two pronged approach handles both weak and strong models.
3. Demonstrate the success of the attack against latest models.

**Weaknesses:**

1. The paper's title is a bit misleading. The proposed attack does not seem adaptive against a defense that knows about the attack.
2. Authors have argued that AJF can successfully evade three types of safeguards: input filtering, internal safeguards, and output filtering. However, the evaluation fails to evaluate the attack along these dimensions. The attack has not been tested against specialized filters such as LlamaGuard or ShieldGemma.
3. While the additional comprehension and decryption tasks decrease the refusal rate, it might impact the quality of the malicious response.
4. The considered baselines are outdated (GCG, GPTFUZZER). The authors should compare their attack against more recent and stronger attacks such as GOAT and TAP.
5. The paper fails to discuss or compare against important related work such as Many-shot jailbreaking by Anil et al, DeepInception by Li et al, and ArtPrompt by xiang et al.

**Questions:**

How will the attack work if the defense is explicitly finetuned for AJF's binary tree syntax?

---

> ### Author Response · Authors · 2025-11-19
> **I would be deeply grateful and honored if you were willing to further discuss or exchange ideas with me.**
>
> I am truly honored to receive your valuable and insightful review. Please allow me to respond to your comments, and I would be deeply grateful and honored if you were willing to further discuss or exchange ideas with me. My detailed responses are provided below.
>
> Regarding Weakness 1:
>     Thank you very much for your insightful comment. Regarding this concern, our viewpoint is that if a defense is designed specifically and exclusively to counter the proposed method, it is possible to replace En_prompt and En_response with alternative encryption mechanisms.
>     However, for all currently available and widely recognized black-box large language models, our attack methodology remains effective. This effectiveness is also supported by the empirical results presented in the experiments section.
>
> Regarding Weakness 2:
>     The reason we did not include LlamaGuard or ShieldGemma in our evaluation is that prior studies have already demonstrated that these models do not provide strong safety guarantees, or that their safety levels are essentially comparable to the models we have tested.
>     For example, in FlipAttack: Jailbreak LLMs via Flipping by Yue Liu, Xiaoxin He, Miao Xiong, Jinlan Fu, Shumin Deng, Yingwei Ma, Jiaheng Zhang, and Bryan Hooi (ICML 2025), cited in our paper as (Liu et al., 2025), the authors report a 98.65% attack success rate on Llama Guard 7B and 100% on Llama Guard 2 8B, while the success rate on GPT-4 is 89.42%.
>     These results indicate that such “safety-enhanced” models are actually less safety than GPT-4. Therefore, we choose the more secure GPT-4 series for evaluation,
>
> Regarding Weakness 3:
>     Thank you for your valuable suggestion. Enhancing the quality of malicious responses is indeed an interesting direction, and we will consider it in our future work. In the appendix of our paper, we provide an example where our method successfully jailbreaks the LLM, and the LLM produces a relatively clear malicious response.
>
> Regarding Weakness 4:
>     The reason we did not include a comparison with GOAT is that the paper reports using a self-selected subset of AdvBench (Zou et al., 2023) and a portion of HarmBench. Since the authors did not disclose which specific questions were used, it is impossible for us to perform a fair and reproducible comparison. Regarding TAP, our baseline FlipAttack (Liu et al., 2025) has already compared against TAP and achieved a higher jailbreak success rate.
>     We believe our method demonstrates comparable or even stronger advantages. For instance, when attacking GPT-4, GOAT reports an 87.5% success rate using its self-selected dataset (including part of AdvBench), whereas our method achieves 99.8% on the full AdvBench dataset (see Table 3).
>     Moreover, we compare against several strong and up-to-date jailbreak methods, including FlipAttack and CodeChameleon (Lv et al., 2024), whose datasets allow fair and controlled comparisons. For example, FlipAttack uses the full AdvBench dataset, enabling a direct comparison with our results (see Table 3). Similarly, our comparison with CodeChameleon is conducted on the same dataset (see Table 1).
>
> Regarding Weakness 5:
>     Most of these methods you mentioned have already been discussed in detail by our baseline FlipAttack (Liu et al., 2025), and under the same evaluation datasets, FlipAttack achieves a higher jailbreak success rate. Therefore, we choose FlipAttack as a more advanced and representative baseline for our comparative analysis.
>
> Response to Question:
>     Your question is very valuable and thought-provoking, and I am honored to share some of my views and discuss them with you.
>     First, Binary tree syntax is not necessarily something that would be used in jailbreak tasks. If an LLM defense method is specifically fine-tuned against AJF’s binary tree syntax, it may sacrifice the usability of the LLM, so I do not think that practical LLM jailbreak defenses would adopt such an approach. Second, to the best of our knowledge, existing LLM defense papers do not claim that they use binary tree syntax for fine-tuning as a defense mechanism.
>     Finally, if a defense is indeed fine-tuned against AJF’s binary tree syntax, the encryption scheme En_prompt in our method can be replaced with other encryption schemes to attempt jailbreaks, which I believe would still be effective.

---

### Meta-Review · Area_Chair_M7Gy · 2025-12-09

**Summary:**

The paper proposes an Adaptive Jailbreak Framework (AJF) based on the comprehension ability of black-box large language models.  Experimental results demonstrate the effectiveness of AJF approach

**Reviewer Concerns:**

Authors have argued that AJF can successfully evade three types of safeguards: input filtering, internal safeguards, and output filtering. However, the evaluation fails to evaluate the attack along these dimensions. The attack has not been tested against specialized filters such as LlamaGuard or ShieldGemma.
While the additional comprehension and decryption tasks decrease the refusal rate, it might impact the quality of the malicious response.
The considered baselines are outdated (GCG, GPTFUZZER). The authors should compare their attack against more recent and stronger attacks such as GOAT and TAP.
The paper fails to discuss or compare against important related work such as Many-shot jailbreaking by Anil et al, DeepInception by Li et al, and ArtPrompt by xiang et al.
The novelty is limited. The core components are just the extentions if the existing work.
The classification of LLMs into Type-I and Type-II is based on a single Caesar cipher test, which is heuristic and may not reflect real comprehension capability.
The attack prompt templates are deterministic and structurally repetitive, which makes them potentially vulnerable to static pattern-based defenses.
The framework is only tested under static conditions and does not assess resilience against adaptive defenses, fine-tuning, or dynamic moderation strategies.
The paper compares AJF with only a limited set of baselines.

**Reviewer Scores:**

The reviewers socre are cosistent and tend to reject the paper.

The author did not adress these questions.

---

### Decision · Program_Chairs · 2026-01-26

Reject